# The burden of antimicrobial resistance among urinary tract isolates of *Escherichia coli* in the United States in 2017

Ian A. Critchley[1]*, Nicole Cotroneo[1], Michael J. Pucci[1], Rodrigo Mendes[2]

**1** Spero Therapeutics, Cambridge, Massachusetts, United States of America, **2** JMI Laboratories, North Liberty, Iowa, United States of America

☯ These authors contributed equally to this work.
* icritchley@sperotherapeutics.com

**Data Availability Statement:** All relevant data are within the manuscript but additional queries can be conducted using the Microbiology Visualization

## Abstract

Urinary tract infections (UTIs) caused by *Escherichia coli* have been historically managed with oral antibiotics including the cephalosporins, fluoroquinolones and trimethoprim-sulfamethoxazole. The use of these agents is being compromised by the increase in extended spectrum β-lactamase (ESBL)-producing organisms, mostly caused by the emergence and clonal expansion of *E. coli* multilocus sequence typing (ST) 131. In addition, ESBL isolates show co-resistance to many of oral agents. Management of UTIs caused by ESBL and fluoroquinolone-resistant organisms is becoming increasingly challenging to treat outside of the hospital setting with clinicians having to resort to intravenous agents. The aim of this study was to assess the prevalence of ESBL phenotypes and genotypes among UTI isolates of *E. coli* collected in the US during 2017 as well as the impact of co-resistance to oral agents such as the fluoroquinolones and trimethoprim-sulfamethoxazole. The national prevalence of ESBL phenotypes of *E. coli* was 15.7% and was geographically distributed across all nine Census regions. Levofloxacin and trimethoprim-sulfamethoxazole-resistance rates were $\geq$ 24% among all isolates and this co-resistance phenotype was considerably higher among isolates showing an ESBL phenotype ($\geq$ 59.2%) and carrying $bla_{CTX-M-15}$ ($\geq$ 69.5%). The agents with the highest potency against UTI isolates of *E. coli*, including ESBL isolates showing cross-resistance across oral agents, were the intravenous carbapenems. The results of this study indicate that new oral options with the spectrum and potency similar to the intravenous carbapenems would address a significant unmet need for the treatment of UTIs in an era of emergence and clonal expansion of ESBL isolates resistant to several classes of antimicrobial agents, including oral options.

## Introduction

Urinary tract infections (UTIs) caused by pathogens such as *Escherichia coli*, the most prevalent UTI pathogen, have been historically managed with oral antibiotics including the

Platform (https://sentry-mvp.jmilabs.com/). This was cited in the data analyses section of the manuscript.

**Funding:** Our respective employers (Spero Therapeutics and JMI Laboratories) did not have any role in the study design and only provided financial support in the form of salaries and research materials. All authors played an equal role in the design and analyses and presentation of the data arising from this surveillance study.

**Competing interests:** Ian Critchley, Nicole Cotroneo and Michael J. Pucci are employees of Spero Therapeutics. Rodrigo Mendes is an employee of JMI Laboratories. This does not alter our adherence to PLOS ONE policies on sharing data and materials.

cephalosporins, trimethoprim-sulfamethoxazole (TMP-SMX) and the fluoroquinolones. Unfortunately, in recent years we have seen the utility of many of these agents being eroded because of widespread use and the subsequent development of resistance [1]. When ciprofloxacin was first introduced in the mid-1980s resistance among UTI isolates of *E. coli* was nonexistent (<1%)[2]. Fluoroquinolone-resistant *E. coli* has progressively increased in the United States from 1.2% in 1998 [2] to 25% in 2012–2014 [3]. Furthermore, resistance to trimethoprim-sulfamethoxazole among UTI isolates of *E. coli* has also increased from 7 to 9% [4] in 1989 to 1992 to 30% in 2009 to 2013 [5].

The increasing prevalence of the extended spectrum β-lactamases (ESBLs) among Gram-negative organisms also seriously compromises the activity of the cephalosporins such as ceftriaxone that is recommended for treatment of pyelonephritis [6] and many of the oral agents used to treat UTIs such as cefuroxime [7]. ESBL-producing *E. coli* pose additional risk factors including longer duration of hospital stay [8]. Of particular concern are the high levels of antimicrobial co-resistance among ESBL-producing organisms that includes many of the oral agents used to treat UTIs [9, 10]. A particular driver of the rapid rise of ESBL-based resistance is the expansion of the ST131-H30 clone of *E. coli* that is well established as a globally disseminated multidrug resistant clone [11, 12]. The ST131 clone is frequently associated with the CTX-M-15 ESBL which is now the most prevalent ESBL in the US and many other countries [13, 14]. In addition, ST131-H30 isolates are uniformly fluoroquinolone-resistant due to conserved replacement mutations in *gyrA* and *parC* [15] which are responsible for the millions fluoroquinolone and cephalosporin-resistant infections being reported globally [16]. Moreover, this clone has also been associated with a higher rate of persistent UTI, adverse outcomes and empiric antimicrobial therapy failure [17, 18].

Surveillance studies with isolates collected from 2009 to 2011 have shown that the only agents that remain highly active against ESBL-producing *E. coli* and other common uropathogens are the intravenous carbapenems because of their inherent stability to β-lactamases other than carbapenemase enzymes [19]. The goal for this study was to assess the prevalence of ESBL phenotypes and genotypes among UTI isolates of *E. coli* collected in the USA in 2017, and the impact of co-resistance to widely used oral agents such as the fluoroquinolones and trimethoprim-sulfamethoxazole. The study also evaluated the activity of the intravenous carbapenems to determine if they remain highly active against ESBL-producing and fluoroquinolone-resistant organisms.

## Materials and methods

### Bacterial isolates

A total of 1831 isolates of *E. coli* were collected from 30 participating medical centers that were geographically distributed among the 9 USA Census divisions in 2017 as part of the SENTRY surveillance platform (JMI Laboratories, North Liberty, IA, USA). The isolates evaluated in this study were collected from patients with urinary tract infections according to defined protocols [20]. The majority of the isolates were isolates from urine (n = 1449) and 132 isolates were isolated from patients with a ureteral (Foley) catheter. The isolates were from both nosocomial and community-acquired infections and included complicated and uncomplicated UTIs. Only isolates determined to be significant by local criteria as the reported probable cause of infection were included in the study. Species identification was confirmed using standard biochemical tests and using a MALDI Biotyper (Bruker Daltronics, Billerica, MA) according to the manufacturer's instructions [21].

## Susceptibility testing

All isolates were centrally tested (JMI Laboratories, North Liberty, IA) using the broth microdilution method in accordance with CLSI guidelines. In particular, the antibiotics evaluated in the study included various oral antibiotics routinely used to treat UTIs including the cephalosporins, fluoroquinolones and trimethoprim-sulfamethoxazole, as well as the intravenous carbapenems and other agents used to treat UTIs. ESBL phenotypes were determined in accordance with CLSI MIC screening criteria, as previously described [22]. CLSI susceptibility interpretive criteria for the Enterobacteriaceae were used to determine susceptibility and resistance rates for all agents where appropriate, including for determining the fluoroquinolone- and trimethoprim-sulfamethoxazole-resistant subsets (CLSI M100-S28) [23]

## Resistant subsets and β-lactamase screening

Isolates were screened *in silico* for narrow- and extended-spectrum β-lactamases, including carbapenemases and isolates that met ESBL MIC screening criteria were further analyzed using molecular methods (Next-Generation Sequencing; NGS) to identify specific β-lactamase genes such as $bla_{CTX-M-15}$. The usual β-lactamase profiles observed in these surveillance studies have been previously published [22, 24, 25]. For NGS, DNA extracts were quantified using the Qubit High Sensitivity DS-DNA assay (Invitrogen/Thermo Fisher, Inc) and normalized to 0.2 ng/µL. A total of 1 ng of high-quality genomic DNA was used as input material for library construction using the Nextera XT DNA library preparation kit (Illumina, San Diego, CA). Libraries were normalized using the bead-based normalization procedure (Illumina) and sequenced on MiSeq. Fastq files generated were assembled using SPAdes Assembler and subjected to proprietary software (JMI Laboratories) for screening of β-lactamase genes [21].

## Data analyses

All data and analysis reported in this study were conducted using the publicly available Microbiology Visualization Platform (https://sentry-mvp.jmilabs.com/). This freely available online tool provides query and analysis capability of the SENTRY Antimicrobial Surveillance Program database and it was used for this study to generate national and regional resistance rates and analyze co-resistance for the ESBL phenotypes as well as the susceptibility results for the $bla_{CTX-M-15}$ genotypes of *E. coli*.

## Results

### Susceptibility of all UTI isolates of *E. coli* collected in the USA during 2017

The results in Table 1 show the susceptibility results for different antimicrobial agents against the 1831 isolates of *E. coli* collected from UTI patients. Resistance to levofloxacin and ciprofloxacin were observed in 24.3% and 25.8% of isolates, respectively. A TMP-SMX resistance phenotype was noted in 32.1% of isolates. Using the oral breakpoints for cefuroxime, 15.9% of isolates were resistsant. In contrast, the intravenous carbapenems including doripenem, ertapenem, imipenem and meropenem were all highly active (≥99.4% susceptible) with little or no resistance being observed. Among other agents, amikacin was also one of the most active agents with only 0.1% of the UTI isolates of *E. coli* being resistant. Ampicillin-sulbactam was among one of the least active agents with 28.7% of the isolates being resistant.

**Table 1. Susceptibility results for 1831 isolates of *E. coli* from urinary tract infections collected in the USA in 2017 (SENTRY Antimicrobial Surveillance Program).**

| Agent | MIC (µg/mL) | | | %S[a] | %I[a] | %R[a] |
|---|---|---|---|---|---|---|
| | Range | 50% | 90% | | | |
| Levofloxacin | ≤0.03–>16 | ≤0.03 | 16 | 74.2 | 1.5 | 24.3 |
| Ciprofloxacin | ≤0.03–>4 | ≤0.03 | >4 | 73.9 | 0.3 | 25.8 |
| Trimethroprim-sulfamethoxazole | ≤0.5–>8 | ≤0.5 | >8 | 67.9 | - | 32.1 |
| Cefuroxime | ≤0.12–>64 | 4 | >64 | 63.2 | 20.9 | 15.9 [b] |
| | | | | 80.3 | 3.8 | 15.9 [c] |
| Amoxicillin-clavulanate | 0.5–>32 | 8 | 16 | 77.9 | 16.4 | 5.8 |
| Ampicillin-sulbactam | ≤0.5–>64 | 8 | 64 | 54.1 | 17.3 | 28.7 |
| Piperacillin-tazobactam | ≤0.06–>128 | 2 | 4 | 97.8 | 1.3 | 0.9 |
| Doripenem | ≤0.06–1 | ≤0.06 | ≤0.06 | 100 | 0.0 | 0.0 |
| Ertapenem | ≤0.008–2 | ≤0.008 | 0.03 | 99.4 | 0.3 | 0.2 |
| Imipenem | ≤0.12–1 | ≤0.12 | 0.25 | 100 | 0.0 | 0.0 |
| Meropenem | ≤0.015–1 | ≤0.015 | 0.03 | 100 | 0.0 | 0.0 |
| Cefepime | ≤0.12–>16 | ≤0.12 | 8 | 88.6 | 2.6 | 8.8 [d] |
| Ceftazidime | 0.03–>32 | 0.25 | 8 | 89.0 | 2.5 | 8.5 |
| Amikacin | 1–>32 | 2 | 4 | 99.7 | 0.2 | 0.1 |
| Gentamicin | 0.25–>16 | 0.5 | >16 | 87.7 | 0.4 | 11.9 |
| Doxycycline | 0.25–>8 | 1 | >8 | 72.4 | 7.9 | 19.7 |
| Minocycline | 0.25–>32 | 1 | 8 | 86.9 | 6.3 | 6.8 |
| Tetracycline | 0.5–>16 | 2 | >16 | 70.2 | 0.1 | 29.7 |

[a]2018 CLSI Interpretive criteria, %S = percent susceptible, %I = percent intermediate, %R = percent resistant

[b]Using oral breakpoints

[c]Using parenteral breakpoints

[d]Intermediate interpreted as susceptible-dose-dependent

## Prevalence of ESBL phenotypes of *E. coli* and co-resistance to widely used oral antimicrobial agents

Fig 1 shows the national and regional prevalence of ESBL phenotypes, levofloxacin-resistant and TMP-SMX-resistant isolates of *E. coli* from UTIs in the USA in 2017. There were 287 (15.7%) out of the 1831 isolates of *E. coli* identified as ESBL phenotypes. ESBL phenotypes were identified among isolates from all US Census regions and ranged from 10.5% in West North Central region to 29.6% in the mid-Atlantic region. The national prevalence of levofloxacin-resistant among *E. coli* from UTIs was 24.3% and ranged from 18% in the Mountain region to 38.1% in the mid-Atlantic region. The national prevalence of TMP-SMX-resistant *E. coli* was 32.1% and ranged from 26.8% in East North Central to 43.5% in the mid-Atlantic. The mid-Atlantic region exhibited the highest burden of resistance among UTI *E. coli* among all the Census regions.

ESBL phenotypes were further analyzed as a subgroup to evaluate the extent of co-resistance to other agents including oral agents widely used to treat UTIs. The results in Fig 2 show the resistance rates among the 287 ESBL phenotypes of *E. coli* from UTIs in the USA during 2017. Not surprisingly, 93.6% of the ESBL phenotypes were resistant to cefuroxime. High resistance rates were also observed for ciprofloxacin and levofloxacin at 71.8% and 67.9%, respectively. There was also high resistance to TMP-SMX with 56.1% of the ESBL phenotypes being resistant. In contrast, the agents with the lowest resistance rates were the intravenous

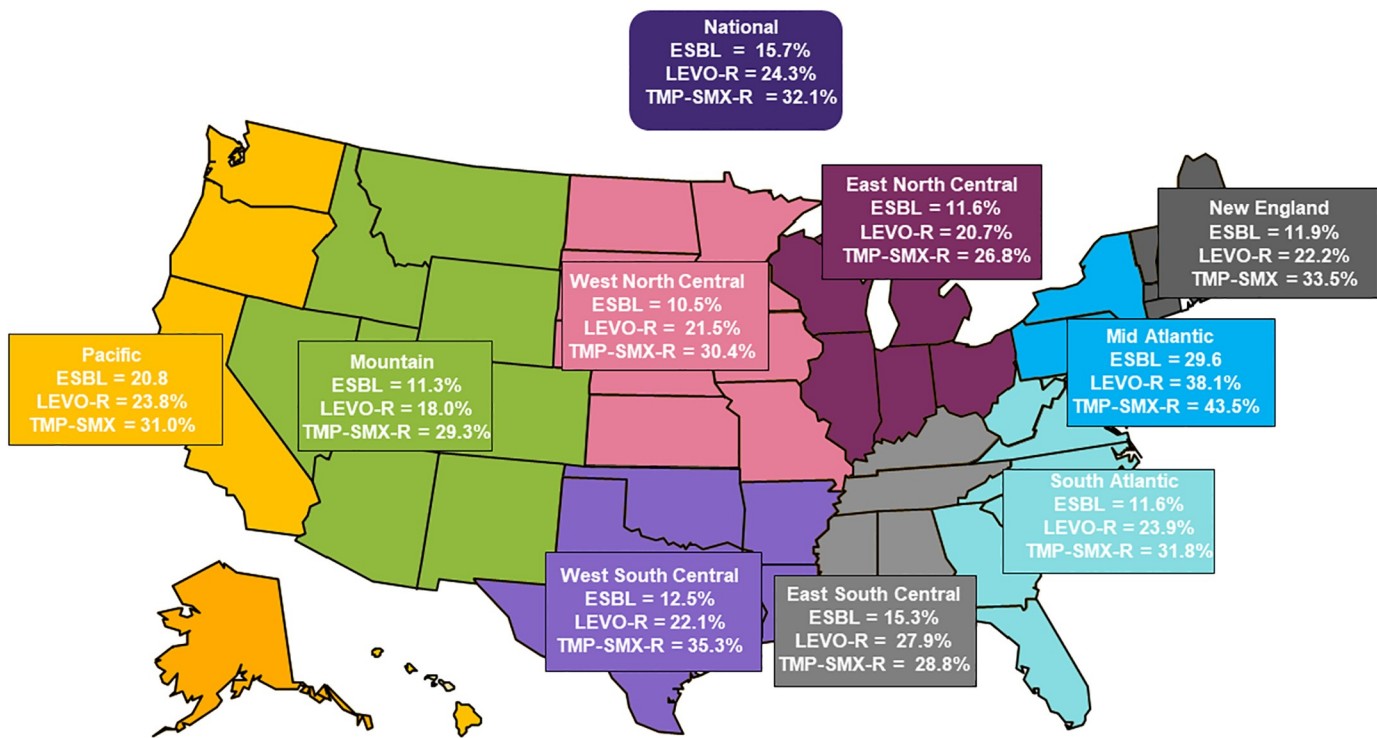

**Fig 1. National and regional prevalence of ESBL phenotypes, levofloxacin- and trimethoprim-sulfamethoxazole-resistant phenotypes among 1831 isolates of *E. coli* from UTIs in the USA in 2017.** ESBL = extended spectrum β-lactamase, LEVO-R = levofloxacin-resistant, TMP-SMX-R = trimethoprim-sulfamethoxazole-resistant.

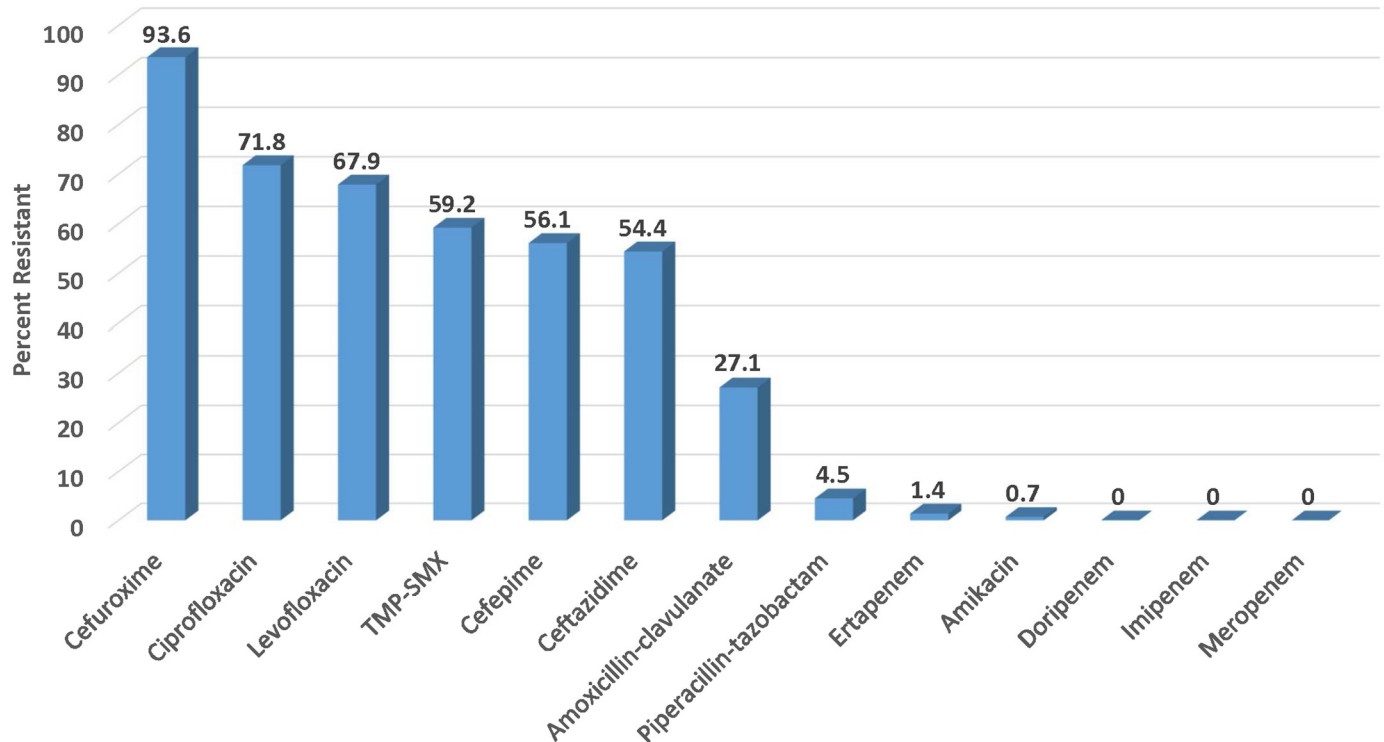

**Fig 2. Antibiotic resistance among 287 ESBL phenotypes of UTI isolates of *E. coli* collected in the USA in 2017.**

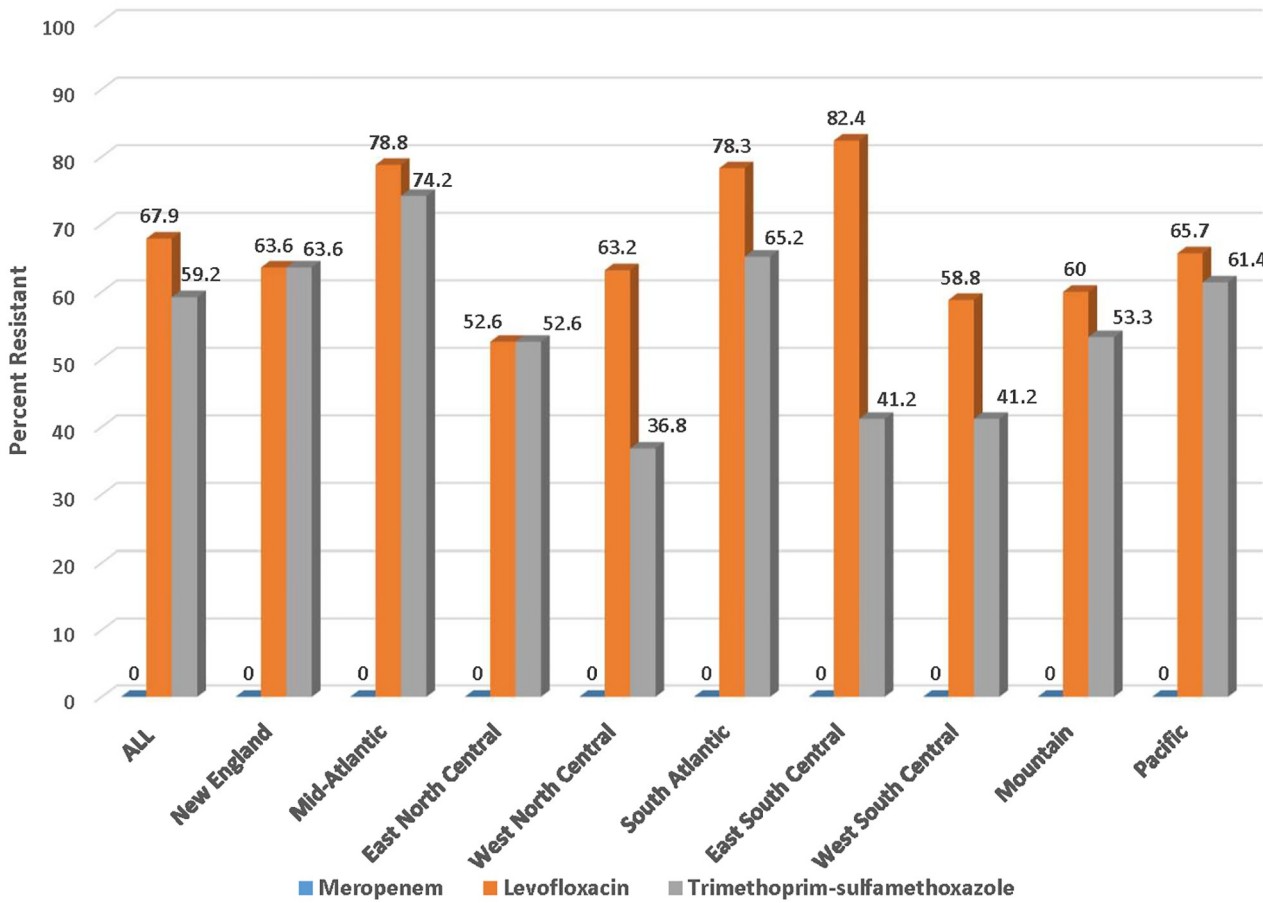

**Fig 3. Resistance to meropenem, levofloxacin and trimethoprim-sulfamethoxazole among 287 ESBL phenotypes of *E. coli* from UTIs in the USA in 2017 according to Census region.**

carbapenems with none of the isolates being resistant to doripenem, imipenem and meropenem and only 1.4% of isolates being resistant to ertapenem.

Fig 3 shows the prevalence of levofloxacin, TMP-SMX and meropenem resistance rates among ESBL isolates of *E. coli* across the 9 Census regions. The national prevalence of levofloxacin-resistance was 67.9% and ranged from 52.6% in East North Central to 82.4% in the East South Central region. Similarly, the national prevalence of TMP-SMX resistance was 59.2% and ranged from 36.8% in West South Central to 74.2% in the mid-Atlantic region. No meropenem-resistant UTI isolates of *E. coli* were identified in any of the Census regions.

## Co-resistance among fluoroquinolone-resistant and TMP-SMX-resistant *E. coli*

The results in Table 2 show the resistance profiles of isolates that are either resistant to levofloxacin or TMP-SMX. Among the levofloxacin-resistant *E. coli* co-resistance was observed for cefuroxime with 45.7% resistance and for TMP-SMX with 56.2% of the isolates being reported as resistant. Similarly, for the TMP-SMX-resistant isolates of *E. coli* 31.3% were co-resistant to cefuroxime and 42.5% co-resistant to levofloxacin. In contrast, little or no resistance was observed for the carbapenems against levofloxacin and/or TMP-SMX-resistant isolates.

**Table 2. Co-resistance among trimethoprim-sulfamethoxazole-resistant and levofloxacin-resistant *E. coli* from urinary tract infections collected in the USA in 2017.**

| Agent | Percent co-resistance among UTI isolates of *E. coli* resistant to: | |
|---|---|---|
| | Trimethoprim-sulfamethoxazole (N = 588) | Levofloxacin (N = 445) |
| Cefuroxime | 31.3 | 45.7 |
| Ceftazidime | 15.0 | 24.7 |
| Ciprofloxacin | 44.2 | 100 |
| Levofloxacin | 42.5 | 100 |
| Doripenem | 0.0 | 0.0 |
| Ertapenem | 0.3 | 0.5 |
| Imipenem | 0.0 | 0.0 |
| Meropenem | 0.0 | 0.0 |
| Trimethoprim-sulfamethoxazole | 100 | 56.2 |

### Susceptibility of *bla*$_{CTX-M-15}$ genotypes of UTI isolates of *E. coli*

The *bla*$_{CTX-M-15}$ genotypes were identified among 151 of the UTI isolates of *E. coli* collected in the USA during 2017 and were the most prevalent ESBLs identified accounting for 59% of the ESBL phenotypes. The next most prevalent β-lactamase was OXA-1/30 but in most isolates was co-expressed among CTX-M-15 positive isolates. The susceptibility results for various antimicrobial agents are shown in Table 3. The isolates were highly resistant to the fluoroquinolones with resistance rates of 81.5% and 83.4%, respectively, for levofloxacin and ciprofloxacin. High resistance was also observed for TMP-SMX (69.5%) and cefuroxime (100%). High resistance rates were also observed for many of the other agents tested with the exception of the carbapenems where none of the isolates were resistant and amikacin where 1.3% of the isolates were resistant.

## Discussion

While oral antibiotics have been a mainstay of therapy for treating UTI's the results from this study show that levofloxacin and/or TMP-SMX resistance rates are ≥ 24% among UTI isolates of *E. coli* collected in the US during 2017. This increase in resistance suggests that considerable caution should be exercised when choosing to use the fluoroquinolones as their widespread use has resulted in being implicated as a "smoking gun" due to their role in promoting resistance [26]. This increase in fluoroquinolone-resistance among UTI isolates of *E. coli* is now resulting in calls to combat their use as first choice agents [27]. In this study fluoroquinolone resistance among UTI isolates of *E. coli* was also geographically distributed across all nine Census regions. The mid-Atlantic region exhibited the highest prevalence of levofloxacin-resistant isolates of *E. coli* (38.1%) and is consistent with prevalence data from other studies [5, 14, 28].

This study also showed the prevalence of ESBL phenotypes of *E. coli* from UTIs was 15.7% and many of these isolates exhibited considerable co-resistance to many of the currently available oral agents. Furthermore, *bla*$_{CTX-M-15}$ was the most prevalent genotype among the UTI isolates of *E. coli* accounting for 59% of the ESBL phenotypes. The increase in prevalence of ESBLs is likely due to the widespread use of the cephalosporins [29]. Another possible factor for the increased prevalence of ESBL phenotypes of *E. coli* is the global dissemination of the ST131 clone that frequently carries *bla*$_{CTX-M-15}$ [30]. In particular, *E. coli* 025b:H4/ST131 is now prevalent in long term care facilities, exhibits co-resistance to the fluoroquinolones, aminoglycosides and TMP-SMX, and now represents a considerable public health concern [11, 12]. The factors responsible for the successful global dissemination of *E. coli* ST131 remain to

**Table 3. Activity of antimicrobial agents against confirmed CTX-M-15 β-lactamase-producing isolates of *E. coli* collected from UTIs in the USA during 2017.**

| Antimicrobial Agent | MIC (µg/mL) | | %S[a] | %I[a] | %R[a] |
|---|---|---|---|---|---|
| | Range | 90% | | | |
| Levofloxacin | ≤0.03–>16 | >16 | 17.2 | 1.3 | 81.5 |
| Ciprofloxacin | ≤0.03–>4 | >4 | 15.2 | 1.3 | 83.4 |
| Trimethoprim-sulfamethoxazole | ≤0.5–>8 | >8 | 30.5 | - | 69.5 |
| Cefuroxime | >64 | >64 | 0.0 | 0.0 | 100[b] |
| | | | 0.0 | 0.0 | 100[c] |
| Amoxicillin-clavulanate | 4–32 | 32 | 34.8 | 52.2 | 13.0 |
| Ampicillin-sulbactam | 4–>64 | 64 | 8.6 | 19.9 | 71.5 |
| Piperacillin-tazobactam | 0.25–>128 | 32 | 89.4 | 6.6 | 4.0 |
| Doripenem | ≤0.06–0.5 | ≤0.06 | 100 | 0.0 | 0.0 |
| Ertapenem | ≤0.008–1 | 0.25 | 97.1 | 2.9 | 0.0 |
| Imipenem | ≤0.12–0.5 | ≤0.12 | 100 | 0.0 | 0.0 |
| Meropenem | ≤0.015–0.5 | 0.06 | 100 | 0.0 | 0.0 |
| Cefepime | 1–>16 | >16 | 9.3 | 9.9 | 80.8[d] |
| Ceftazidime | 1–>32 | >32 | 14.6 | 13.2 | 72.2 |
| Amikacin | 1–>32 | 8 | 96.7 | 2.0 | 1.3 |
| Gentamicin | 0.5–>16 | >16 | 57.0 | 0.0 | 43.0 |
| Doxycycline | 0.5–>8 | >8 | 37.1 | 18.6 | 44.3 |
| Minocycline | 0.5–>32 | 16 | 74.3 | 8.6 | 17.1 |
| Tetracycline | 1–>16 | >16 | 32.9 | 0.0 | 67.1 |

[a]2018 CLSI Interpretive criteria; %S = percent susceptible, %I = percent intermediate, %R = percent resistant

[b]Using oral breakpoints

[c]Using parenteral breakpoints

[d]Intermediate interpreted as susceptible-dose-dependent

be elucidated but may be due to the type I fimbrial adhesins that may allow it to colonize the gastrointestinal tract more efficiently [31–33].

The ESBL phenotypes of *E. coli* reported in this study were geographically distributed across the nine Census regions with the highest prevalence being among isolates collected in the mid-Atlantic region and is similar to the high prevalence reported in previous studies [14]. To further evaluate the co-resistance among ESBL phenotypes, the resistance rates were determined for currently available oral agents that included the fluoroquinolones and TMP-SMX and the high levels of co-resistance at ≥59% have confirmed that high rates of co-resistance exist for contemporary isolates collected in 2017. Furthermore, the increased resistance to TMP-SMX is equally concerning since this resistance is plasmid-mediated with genes that not only encode enzymes such as type II dihydrofolate reductase but also additional genes that confer resistance to other antibiotic classes including the fluoroquinolones with the ability to spread between organisms [34].

This study not only assessed co-resistance among ESBL phenotypes but also among TMP-SMX and fluoroquinolone-resistant isolates. Not surprisingly, the TMP-SMX-resistant isolates of *E. coli* exhibited high co-resistance (≥ 30%) to the fluoroquinolones and cefuroxime. Also, the fluoroquinolone-resistant isolates of *E. coli* exhibited high co-resistance (≥ 45%) to TMP-SMX and ceforuxime. The high co-resistance among the currently available oral agents suggests that, if you lose susceptibility to one, you lose them all.

The increase in resistance to many of the currently available oral options makes the management of UTIs caused by coresistant ESBL-producing organisms a significant challenge for

the clinician to treat outside of the hospital setting. Fosfomycin is one oral option that has seen increased use for the treatment of uncomplicated UTIs but was not tested in this study because of the requirement for testing by agar dilution and represents a limitation of the current study. Current methods to identify fosfomycin-resistant *E. coli* in urine can give very different results highlighting a need to more accurately defined rates of resistance as fosfomycin use increases [35]. While nitrofurantoin is another oral option for uncomplicated UTIs and is active against *E. coli* it is less active against *K. pneumoniae* and *P. mirbilis* that are also implicated in complicated UTI's [36]. Although the results of this study show that the intravenous carbapenems remain very active against most UTI isolates of *E. coli* with little or no carabapenem resistance, no oral options with the spectrum and potency of the carbapenems are currently available. While the development of new and systemic agents have been directed to the treatment of carbapenem-resistant Enterobacteriaceae UTIs, little or no effort has been dedicated to the development of new oral options for the treatment of UTIs caused by ESBL-producing and fluoroquinolone-resistant organisms. The development of new oral options present additional challenges, since they must be stable in solid form, and possess the appropriate pharmacodynamic properties once adequately dissolved and adsorbed in the GI tract, these agents need to reach the site of infection. Oral agents with the spectrum and potency of the intravenous carbapenems would address a substantial unmet need for new options to treat multi-drug-resistant pathogens implicated in complicated UTIs. In particular, the carbapenems are inherently stable to the ESBL and Class C (AmpC) β-lactamases, present in organisms that are prevalent among common Gram-negative UTI pathogens [37].

## Author Contributions

**Conceptualization:** Ian A. Critchley, Nicole Cotroneo, Michael J. Pucci, Rodrigo Mendes.

**Data curation:** Ian A. Critchley.

**Validation:** Rodrigo Mendes.

**Writing – original draft:** Ian A. Critchley, Rodrigo Mendes.

**Writing – review & editing:** Nicole Cotroneo, Michael J. Pucci, Rodrigo Mendes.

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
