## [Decision Letter · Decision Letter 0]

13 Nov 2019

PONE-D-19-18189

The Burden of Antimicrobial Resistance among Urinary Tract Isolates of Escherichia coli in the United States in 2017

PLOS ONE

Dear Dr Critchley ,

Thank you for submitting your manuscript to PLOS ONE. The paper is well-written and easy to follow. It provides useful information to clinicians in the USA and demonstrates the urgent need for additional oral antibiotics the treatment of urinary tract infections caused by multidrug-resistant Enterobacteriaceae. After careful consideration, we feel that it has merit but does not fully meet PLOS ONE’s publication criteria as it currently stands. Therefore, we invite you to submit a revised version of the manuscript that addresses the points raised during the review process.

Minor suggestions / recommendations:

Page 5 Bacterial isolates: If this information is available, it would be useful to state what proportion of isolates were from community-acquired infections and what proportion were from hospital-acquired infections; also, from which anatomical sites were these organisms isolated? If this is not possible, state that these are from both settings and/or that sites are unknown, respectively. Reference 20 is not clear regarding this issue.

Page 5 Susceptibility testing: Add the CLSI M100-S28 in the references

Page 5 Resistance subsets: Consider stating which beta-lactamase genes in addition to blaCTX-M-15 were screened for. Furthermore, further characterisation of a common genotype, blaCTX-M15, could   explain the persistence of these specific isolates in causing infections.    

Page 7 Susceptibility of all UTI isolates: Use the term "resistance/resistant" with the appropriate % consistently rather than interchanging it with "non-susceptible".

Page 7 Prevalence of ESBL phenotypes Figure 2: Consider adding amikacin to figure 2. The high rate of amikacin susceptibility in the ESBL subset provides a carbapenem-sparing treatment option.

Page 8 Figure 3: Showing ertapenem resistance rates would yield more useful information than meropenem does. It is clear that all isolates (ESBL and non-ESBL) are meropenem susceptible from Table 1. Ertapenem is a narrower spectrum carbapenem and where possible should be used preferentially.

Page 8 Co-resistance among fluoroquinolone-resistant Table 2: Consider adding the amoxicillin-clavulanate data.

Page 8 Susceptibility of blaCTX-M-15 genotypes: State what % the 151 represents of the total ESBL isolates. If any other predominating beta-lactamase type was found, add this information. If not, state this.

Page 8 Susceptibility of blaCTX-M-15 genotypes Table 3: Correct year in title of Table 3 (2017 rather than 2016).

Page 8 Susceptibility of blaCTX-M-15 genotypes Line 221: Should read "of the carbapenems" not "carbapenem".

Page 9 Paragraph 2: Comment on the % blaCTX-M-15 found.

Page 10: Nitrofurantoin and fosfomycin data are not presented-add this as a limitation of the study. Authors discuss the need for new oral antibiotics particularly from carbapenem class but perhaps testing of fosfomycin would have shown susceptibility among these pathogens. Fosfomycin is an old antibiotic, used as an oral treatment for uncomplicated urinary tract infections. Fosfomycin has been shown to have activity against some resistant uropathogens suggesting that this antibiotic may provide a useful option for the treatment of patients with difficult-to-treat-infections.

We would appreciate receiving your revised manuscript by 30 November 2019. To enhance the reproducibility of your results, we recommend that if applicable you deposit your laboratory protocols in protocols.io, where a protocol can be assigned its own identifier (DOI) such that it can be cited independently in the future. For instructions see: http://journals.plos.org/plosone/s/submission-guidelines#loc-laboratory-protocols

We look forward to receiving your revised manuscript.

Kind regards,

Adriano Gianmaria Duse, MD

Academic Editor

PLOS ONE

1. We noticed you have some minor occurrence(s) of overlapping text with the following previous publication(s), which needs to be addressed:

https://aac.asm.org/content/61/11/e01045-17

In your revision ensure you cite all your sources (including your own works), and quote or rephrase any duplicated text outside the Methods section. Further consideration is dependent on these concerns being addressed.

2. Thank you for including your competing interests statement; "Ian Critchley, Nicole Cotroneo and Michael J. Pucci are employees of Spero Therapeutics. Rodrigo Mendes is an employee of JMI Laboratories"

We note that one or more of the authors are employed by a commercial company: name of commercial company.

---

## [Author Response · Author response to Decision Letter 0]

22 Nov 2019

Response to reviewer comments:

Page 5 Bacterial isolates: If this information is available, it would be useful to state what proportion of isolates were from community-acquired infections and what proportion were from hospital-acquired infections; also, from which anatomical sites were these organisms isolated? If this is not possible, state that these are from both settings and/or that sites are unknown, respectively. Reference 20 is not clear regarding this issue.

Authors’ response: The isolates that were tested represent both nosocomial and community-acquired infections from both uncomplicated and complicated UTIs. Unfortunately, we do not have access to enough demographic data to be able to separate isolates from community versus hospital-acquired infections. We have added information on anatomical sites with the majority of the organisms being isolated from a urine culture and 132 isolates being isolates from patients with the Foley catheter.

Page 5 Susceptibility testing: Add the CLSI M100-S28 in the references

Authors’ response: This has been added to the reference list

Page 5 Resistance subsets: Consider stating which beta-lactamase genes in addition to blaCTX-M-15 were screened for. Furthermore, further characterisation of a common genotype, blaCTX-M15, could explain the persistence of these specific isolates in causing infections.

Authors’ response: The isolates were screened in silico for narrow and extended spectrum β-lactamases, including carbapenemases. We have also added a comment stating that the usual β -lactamase profiles observed in these studies have been previously published (references added). [Mendes et al., 2019 doi: 10.1093/ofid/ofz004, Castenheira et al., 2019 doi: 10.1128/AAC.00160-19 and Mendes et al., 2019 doi: 10.1089/mdr.2019.0198].

Page 7 Susceptibility of all UTI isolates: Use the term "resistance/resistant" with the appropriate % consistently rather than interchanging it with "non-susceptible".

Authors’ response: Changes made to substitute the non-susceptible rates with resistance rates

Page 7 Prevalence of ESBL phenotypes Figure 2: Consider adding amikacin to figure 2. The high rate of amikacin susceptibility in the ESBL subset provides a carbapenem-sparing treatment option.

Authors’ response: Figure 2 has been amended to include amikacin.

Page 8 Figure 3: Showing ertapenem resistance rates would yield more useful information than meropenem does. It is clear that all isolates (ESBL and non-ESBL) are meropenem susceptible from Table 1. Ertapenem is a narrower spectrum carbapenem and where possible should be used preferentially.

Authors’ response: Yes, we agree but unfortunately not all the 287 ESBL phenotypes of E. coli were tested against ertapenem (only 141 isolates had available susceptibility results for ertapenem) so we selected meropenem as a representative carbapenem since data was available for all 287 ESBL phenotypes for these three drugs.

Page 8 Co-resistance among fluoroquinolone-resistant Table 2: Consider adding the amoxicillin-clavulanate data.

Authors’ response: Although only 8.6% of the levofloxacin-resistant isolates were resistant to amoxicillin-clavulanate, 34.3% of the isolates displayed were in the intermediate category that would not be reflected in the current presentation of the table and was a factor in our reason not to include. 

Page 8 Susceptibility of blaCTX-M-15 genotypes: State what % the 151 represents of the total ESBL isolates. If any other predominating beta-lactamase type was found, add this information. If not, state this.

Authors’ response: This been added to the text to highlight that CTX-M-15 was the most prevalent accounting for 59% of the ESBL phenotypes of E. coli. Also highlighted that OXA-1/30 was the next most prevalent but in most isolates was co-expressed with CTX-M-15.

Page 8 Susceptibility of blaCTX-M-15 genotypes Table 3: Correct year in title of Table 3 (2017 rather than 2016).

Authors’ response: This has now been corrected to reflect 2017

Page 8 Susceptibility of blaCTX-M-15 genotypes Line 221: Should read "of the carbapenems" not "carbapenem".

Authors’ response: This has been corrected

Page 9 Paragraph 2: Comment on the % blaCTX-M-15 found.

Authors’ response: Added the following text: “Furthermore, blaCTX-M-15 was the most prevalent genotype among the UTI isolates of E. coli accounting for 59% of all the ESBL phenotypes

Page 10: Nitrofurantoin and fosfomycin data are not presented-add this as a limitation of the study. Authors discuss the need for new oral antibiotics particularly from carbapenem class but perhaps testing of fosfomycin would have shown susceptibility among these pathogens. Fosfomycin is an old antibiotic, used as an oral treatment for uncomplicated urinary tract infections. Fosfomycin has been shown to have activity against some resistant uropathogens suggesting that this antibiotic may provide a useful option for the treatment of patients with difficult-to-treat-infections.

Authors’ response: We have added some text to the discussion regarding fosfomycin and nitrofurantoin as oral options for uncomplicated UTI. We included an explanation of why fosfomycin was not tested because of the requirement for agar dilution and therefore a limitation of the current study. Also included a sentence on nitrofurantoin and although a good option of uUTIs caused by E. coli it is not suitable for complicated UTIs where other pathogens such as K. pneumoniae and P. mirabilis are not well covered with this agent.

Additional Author Response to Reviewer Questions:

1. We noticed you have some minor occurrence(s) of overlapping text with the following previous publication(s), which needs to be addressed:

https://aac.asm.org/content/61/11/e01045-17

In your revision ensure you cite all your sources (including your own works), and quote or rephrase any duplicated text outside the Methods section. Further consideration is dependent on these concerns being addressed.

Authors’ response: All the overlapping text is from the methods section only and there is no duplicated text from outside the Methods section. Both studies were conducted in the same laboratory and used the same methods to molecularly characterize the isolates. To make it clear we have added a reference citation in the methods section to the article by Sader et al. (https://aac.asm.org/content/61/11/e01045-17) to reference the methods used in that study.

2. Thank you for including your competing interests statement; "Ian Critchley, Nicole Cotroneo and Michael J. Pucci are employees of Spero Therapeutics. Rodrigo Mendes is an employee of JMI Laboratories"

We note that one or more of the authors are employed by a commercial company: name of commercial company.

Authors’ response: The companies listed above are still relevant as described above for all the authors

Authors’ response: Our respective employers (Spero Therapeutics and JMI Laboratories) did not have any role in the study design and only provided financial support in the form of salaries and research materials. All authors played an equal role in the design and analyses and presentation of the data arising from this surveillance study. 

Authors' response: We confirm our adherence to all PLOS ONE policies on sharing data and materials and provide the following statement "This does not alter our adherence to PLOS ONE policies on sharing data and materials." (as detailed in your guide for authors http://journals.plos.org/plosone/s/competing-interests)

---

## [Editor Report · Decision Letter 1]

26 Nov 2019

The burden of antimicrobial resistance among urinary tract isolates of Escherichia coli in the United States in 2017

PONE-D-19-18189R1

Dear Dr. Critchley

We are pleased to inform you that your manuscript has been judged scientifically suitable for publication and will be formally accepted for publication once it complies with all outstanding technical requirements.

With kind regards,

Adriano Gianmaria Duse, MD

Academic Editor

PLOS ONE
---

## [Editor Report · Acceptance letter]

2 Dec 2019

PONE-D-19-18189R1 

The burden of antimicrobial resistance among urinary tract isolates of *Escherichia coli* in the United States in 2017 

Dear Dr. Critchley:

I am pleased to inform you that your manuscript has been deemed suitable for publication in PLOS ONE. Congratulations! Your manuscript is now with our production department. 

With kind regards,

on behalf of

Dr. Adriano Gianmaria Duse 

Academic Editor

PLOS ONE